# Changes in HIV incidence during the COVID-19 pandemic (2020–22) compared with the pre-pandemic period (2015–19) in Peru: An observational study

**Max Carlos Ramírez-Soto**[1]* and **Hugo Arroyo-Hernández**[2]

**1** Facultad de Ciencias de la Salud, Universidad Tecnológica del Peru, Lima, Peru, **2** Universidad de Huánuco, Huánuco, Peru

* maxcrs22@gmail.com

## Abstract

### Introduction

During the COVID-19 pandemic, non-pharmaceutical interventions affected the screening of sexually transmitted infections. We investigated the incidence of HIV infection during the COVID-19 pandemic compared with incidence in the pre-pandemic period.

### Methods

In this observational study, we analyzed HIV surveillance data for all age-groups from 25 geographically diverse regions in Peru from Jan 1, 2015 to Dec 31, 2022. HIV incidence during the COVID-19 pandemic (2020, 2021, and 2022) was compared with pre-pandemic rates (2015–19).

### Results

Overall, there were 65,166 new cases of HIV infection from January 1, 2015 to December 31, 2022. HIV incidence risk ratio (IRR) was 26% lower in 2020 (IRR = 0.74; 95% CI, 0.71–0.76), 5% higher in 2021 (IRR = 1.05; 95% CI, 1.02–1.08) and 16% higher in 2022 (IRR = 1.16; CI, 1.13–1.20), compared with the pre-pandemic period. Furthermore, compared with the pre-pandemic period, the annual incidence of HIV among men was 29% lower in 2020 (IRR = 0.71; 95% CI, 0.68–0.73), 4% higher in 2021 (IRR = 1.04, 95% CI, 1.01–1.08) and 10% higher in 2022 (IRR = 1.10; 95% CI, 1.06–1.14). In the age-stratified analysis, the annual HIV incidence in 2020 was 21 and 33% lower for those aged 18–29 (IRR = 0.79; 95% CI, 0.75–0.83) and 30–59 (IRR = 0.67; 95% CI, 0.64–0.70), respectively, compared with the pre-pandemic period. Finally, annual HIV incidence has decreased in 11 out of 25 regions in 2020, compared with the pre-pandemic period.

**Data availability statement:** The HIV case data used in our analyses are publicly available on the website of Peru's Centro Nacional de Epidemiología, Prevención y Control de Enfermedades (https://www.dge. gob.pe/vih/#grafico02). Third-party Peruvian population figures for this study are publicly available from the MINSA website (https://www. minsa.gob.pe/reunis/?op=1&niv=5&tbl=1). The authors confirm that the data supporting the findings of this study are available from these third-party sources and within the paper.

**Funding:** This study was financially supported by Universidad Tecnológica del Perú in the form of an award for article processing charges. No additional external funding was received for this study.

**Competing interests:** The authors have declared that no competing interests exist.

## Conclusions

Our study showed that during the COVID-19 pandemic in 2020, the incidence of HIV infection in the population of Peru decreased. However, this incidence began to return to pre-pandemic rates in 2021, coinciding with the easing or elimination of non-pharmaceutical interventions. By 2022, the incidence of HIV infection was higher than in the pre-pandemic period, especially in regions of the Peruvian Amazon.

## 1. Introduction

During the COVID-19 pandemic, non-pharmaceutical interventions such as social distancing, travel ban (local or international), movement limitation, restriction of assembly, quarantine for travelers, school closure, workplace closure, and contact tracing were implemented to reduce and control transmission of SARS-CoV-2 [1,2]. As a result, several changes to the health system have had a significant impact on healthcare. Many outpatient services were limited or suspended to prevent the spread of the virus and to conserve resources for COVID-19 care [3–5]. As a result, routine check-ups, elective procedures and some specialist services were postponed or canceled. Many healthcare providers turned to telemedicine to continue providing care while minimizing face-to-face contact. This allowed consultations, follow-ups and even some diagnostics to be carried out remotely [6]. However, it also highlighted inequalities in access to technology and reliable internet. Laboratories had to prioritize on COVID-19 testing, resulting in reduced availability of other diagnostic tests. Many tests and services that were not related to COVID-19 were deprioritized, which had an impact on the monitoring and management of chronic diseases and other health issues [7]. Clinics and hospitals restructured their physical spaces to improve their infection control measures. This included creating separate areas for COVID-19 patients, reconfiguring waiting rooms to promote social distancing, and implementing stricter hygiene protocols. Fear of COVID-19 infection also led many patients to delay or avoid medical care altogether [6,8].

The disruption to health services caused by the COVID-19 pandemic has had a notable and worrying impact on other areas of public health, particularly in the area of HIV testing and diagnosis [7,9,10]. In several countries, there has been a significant decline in HIV testing as a result of the diversion of health care resources to the COVID-19 response [11,12]. In Belgium, there was a 50% decrease in HIV tests [13]. In Australia, testing decreased by 31% in a large STI clinic [14]. In China, a 59% reduction in testing was observed across four regions [15]. In Kenya, challenges included a reduction in clinic-based testing and limited distribution of self-testing kits [16]. A study in Europe reported a drop of more than 50% in HIV testing from March to May 2020, with a slight recovery in the following months [17]. In Japan, a similar decline in HIV testing has been observed [18].

To our knowledge, only two studies have assessed changes in HIV diagnoses in Peru [19,20]. A study found that there was a decrease (33%) in the average number of HIV diagnoses in 2020 (n = 5170), compared to the average number of cases

between 2017 and 2019 (n = 7746). This decrease in the average number of HIV diagnoses was observed in 21 out of 25 regions [19]. Although HIV occurs mainly in men aged 30 and over, the study did not analyze changes in HIV diagnosis by sex and age. It also did not analyze whether changes in HIV diagnoses were sustained in 2021 and 2022 (pandemic period) [19]. Another study found a mean 8.33% (95% CI: −10.73% to −5.93%) decrease in HIV testing proportion after the COVID-19 lockdown, compared with the pre-pandemic period. Additionally, HIV testing rates decreased in 23 of 25 regions [20].

Delays in HIV testing availability have important implications for the timely initiation of treatment and linkage to care. In addition, undiagnosed and untreated HIV increases the risk of transmission, potentially exacerbating the epidemic. Therefore, knowing the impact of the COVID-19 pandemic on the detection of new HIV cases will help guide public health resources to reactivate HIV prevention and testing efforts. In addition, given the multiple "waves" of COVID-19 in Peru, a longer-term assessment of whether subsequent changes in restrictions have had an impact on HIV testing and whether or not testing rates are recovering is warranted.

To better understand how COVID-19-related restrictive measures affected HIV care systems in Peru—and to be better prepared for similar situations in the future—we evaluated changes in the number of reported HIV cases and incidence rates during the COVID-19 pandemic years (2020, 2021 and 2022) compared with the pre-pandemic period (2015−19).

## 2. Methods

### Study design and data sources

We performed an observational study following the Strengthening the Reporting of Observational Studies in Epidemiology reporting guidelines (**STROBE,** S1 Table) [21]. In this observational study, we included HIV surveillance data from 1 January 2015 to 31 December 2022 from all 25 regions of Peru that are administrative departments.

Data on the epidemiological surveillance of HIV in Peru were obtained from Centro Nacional de Epidemiología, Prevención y Control de Diseases, Ministerio de Salud del Peru (website: https://www.dge.gob.pe/vih/#grafico02) [22]. HIV surveillance data included official on new confirmed cases. Data collected included the number of HIV cases by age group (0–11, 12–17, 18–29, 30–59 and ≥60 years), sex (male and female) and geographical region for each year. No other information was available from Centro Nacional de Epidemiología, Prevención y Control de Diseases website.

### Outcome

The changes in the reporting of HIV cases (in terms of numbers and percentages) during the COVID-19 pandemic (in 2020, 2021 and 2022) were the primary outcome. The incidence of HIV infection before and during the COVID-19 pandemic was assessed as a secondary outcome.

### Statistical analyses

We estimated the changes in HIV case reporting (in numbers and percentages) in the COVID-19 pandemic years (2020, 2021 and 2022), by sex and age, and in each geographical region. The average number of HIV cases in the years 2015–2019 was used to calculate the number of HIV cases in the pre-pandemic period. This average number of HIV cases by sex, age and region in the 5 years before the pandemic was the reference number for the comparison with the pandemic years of COVID-19 (2020, 2021 and 2022). Changes in HIV case notification during the pandemic period were estimated as the difference between HIV cases observed in 2020, 2021 and 2022 and the average number of HIV cases reported between 2015 and 2019. Changes in the proportion of HIV cases reported during the pandemic were calculated as: ((number of HIV cases reported in 2020, 2021 or 2022 – average number of HIV cases in the years 2015–2019)/average number of HIV cases in the years 2015–2019 × 100%) by sex, age and region.

The yearly crude incidence rate (IR) for HIV per 100 000 inhabitants was calculated as the number of HIV cases divided by the yearly population estimates (by sex, age group or region) and multiplied by 100 000. Exact 95% CIs were calculated for all IRs assuming a Poisson distribution. Age-, sex- and region-specific incidence rates between 2015 and 2019 were calculated as the total number of HIV cases during the period divided by the sum of the annual population estimates for the 5 years, expressed per 100 000 inhabitants (by sex, age group or region). IR ratios (IRRs) with 95% CIs were calculated for HIV cases per sex, age and region after comparing annual HIV incidence rates during the COVID-19 pandemic period (2020, 2021 and 2022) with the pooled pre-pandemic incidence as the unexposed group. The yearly population figures by region used for the calculation of the IR were obtained from the projections of the National Institute of Statistics and Informatics of Peru. Statistical analyses were performed with the StataSE 17.0 software.

### Ethics

This study was approved (Ethics code: 112–2025-CEI/UTP) by the Institutional Review Board of Universidad Tecnologica del Peru (Lima, Peru). Informed consent was not required due to the observational nature of the study. Furthermore, all data sets were aggregated data and therefore completely anonymous.

## 3. Results

From 1 January 2015 to 31 December 2022, there were 65,166 new cases of HIV infection. In the first year of the COVID-19 pandemic (2020), there was a decrease (26%) in the number of HIV cases (n = 6025), compared with the yearly average number of cases between 2015 and 2019 (n = 8141). In the second and third years of the COVID-19 pandemic (2021 and 2022), there was an increased number of HIV cases (6.7% and 19.8%, respectively), compared with the yearly average number of cases between 2015 and 2019 (Table 1). In addition, in the first year of the COVID-19 pandemic (2020), the number of HIV cases decreased in 21 out of 25 regions compared with the average number of cases between 2015 and 2019. In the second and third years of the pandemic (2021 and 2022), the number of HIV cases decreased in 7 and 5 regions, respectively, compared with the average number of cases between 2015 and 2019 (Table 1).

The annual incidence of HIV infection was similar from 2015 to 2019, with minimal annual variation by sex and age (S2 Table). Compared with the pre-pandemic period overall incidence of 25.03 (95% CI, 24.48–25.58) per 100 000 habitants, the HIV overall incidence was 26% lower in 2020 (IRR = 0.74; 95% CI, 0.71–0.76) and 5% higher in 2021 (IRR = 1.05; 95% CI, 1.02–1.08). By 2022, HIV overall incidence was higher than during the pre-pandemic period (IR = 29.19; 95% CI, 28.61–29.78; Table 2).

Compared with pre-pandemic period overall HIV incidence among men (IR = 39.57; 95% CI, 38.60–40.55), the incidence was 29% lower in 2020 (IRR = 0.71; 95% CI, 0.68–0.73), but 4% higher in 2021 (IRR = 1.04; 95% CI, 1.01–1.08) and 10% higher in 2022 (IRR = 1.10; 95% CI, 1.06–1.14) (Table 2).

In the age-group stratified analysis, in comparison with the pre-pandemic period, the annual HIV incidence decreased in 2020 for all age groups, mainly for the 18–29 (IR = 42.94; 95% CI, 41.36–44.58) and 30–59 (IR = 22.14; 95% CI, 21.32–22.98). This decrease was 21 and 33% greater in 2020 for those aged 18–29 (IRR = 0.79; 95% CI, 0.75–0.83) and 30–59 (IRR = 0.67; 95% CI, 0.64–0.70), respectively, compared to the pre-pandemic period. There was also a higher incidence for the years 2021 and 2022. Similar trends were observed for annual HIV incidence in the stratified analysis by age group compared with the overall analysis in men, with decreases of 39% and 46% in 2020 for those aged 18–29 years (IRR = 0.61; 95% CI, 0.57–0.64) and 30–59 years (IRR = 0.54; 95% CI, 0.51–0.56), respectively (Table 2).

The annual incidence of HIV infection among regions was similar from 2015 to 2019, with minimal annual variation (S3 Table). Compared with the pre-pandemic period, annual HIV incidence decreased in 11 out of 25 regions in 2020. In the regions of Amazonas, Callao, Loreto, Madre de Dios and Ucayali, where the annual HIV incidence was the highest in the pre-pandemic period, this incidence was 45% (IRR = 0.55; 95% CI, 0.44–0.69), 47% (IRR = 0.53; 95% CI, 0.47–0.60), 33% (IRR = 0.67; 95% CI, 0.59–0.77), 19% (IRR = 0.81; 95% CI, 0.5846–1.13) and 25% (IRR = 0.75; 95% CI, 0.64–0.88)

**Table 1. HIV cases before and during the COVID-19 pandemic in Peru.**

| | Pre-pandemic (2015–19) | 2020 | | | 2021 | | | 2022 | | |
|---|---|---|---|---|---|---|---|---|---|---|
| | Mean (cases) | n | Differences in the number of cases | (%) | n | Differences in the number of cases | (%) | n | Differences in the number of cases | (%) |
| **Total** | 8141 | 6025 | −2116 | −26.0 | 8684 | 543 | 6.7 | 9750 | 1609 | 19.8 |
| **Sex** | | | | | | | | | | |
| Male | 6336 | 4650 | −1686 | −26.6 | 6997 | 661 | 10.4 | 7808 | 1472 | 23.2 |
| Female | 1805 | 1375 | −430 | −23.8 | 1687 | −118 | −6.5 | 1942 | 137 | 7.6 |
| **Age group (years)** | | | | | | | | | | |
| 0–11 | 94 | 36 | −58 | −61.6 | 44 | −50 | −53.1 | 68 | −26 | −27.5 |
| 12–17 | 162 | 128 | −34 | −21.2 | 191 | 29 | 17.6 | 298 | 136 | 83.5 |
| 18–29 | 3659 | 2818 | −841 | −23.0 | 4008 | 349 | 9.5 | 4364 | 705 | 19.3 |
| 30–59 | 3960 | 2862 | −1098 | −27.7 | 4184 | 224 | 5.7 | 4716 | 756 | 19.1 |
| ≥60 | 265 | 181 | −84 | −31.7 | 257 | −8 | −3.1 | 304 | 39 | 14.6 |
| **Age group in men (years)** | | | | | | | | | | |
| 0–11 | 54 | 20 | −34 | −63.0 | 25 | −29 | −53.7 | 42 | −12 | −22.2 |
| 12–17 | 96 | 67 | −29 | −29.9 | 111 | 15 | 16.1 | 178 | 82 | 86.2 |
| 18–29 | 2942 | 2225 | −717 | −24.4 | 3323 | 381 | 13.0 | 3632 | 690 | 23.5 |
| 30–59 | 3038 | 2200 | −838 | −27.6 | 3338 | 300 | 9.9 | 3715 | 677 | 22.3 |
| ≥60 | 206 | 138 | −68 | −33.0 | 200 | −6 | −2.9 | 241 | 35 | 17.0 |
| **Age group in women (years)** | | | | | | | | | | |
| 0–11 | 40 | 16 | −24 | −59.8 | 19 | −21 | −52.3 | 26 | −14 | −34.7 |
| 12–17 | 67 | 61 | −6 | −8.7 | 80 | 13 | 19.8 | 120 | 53 | 79.6 |
| 18–29 | 717 | 593 | −124 | −17.3 | 685 | −32 | −4.5 | 732 | 15 | 2.1 |
| 30–59 | 922 | 662 | −260 | −28.2 | 846 | −76 | −8.2 | 1001 | 79 | 8.6 |
| ≥60 | 59 | 43 | −16 | −27.4 | 57 | −2 | −3.7 | 63 | 4 | 6.4 |
| **Region** | | | | | | | | | | |
| Amazonas | 221 | 125 | −96 | −43.5 | 159 | −62 | −28.2 | 384 | 163 | 73.4 |
| Ancash | 137 | 87 | −50 | −36.7 | 174 | 37 | 26.6 | 196 | 59 | 42.6 |
| Apurimac | 10 | 13 | 3 | 32.7 | 25 | 15 | 155.1 | 29 | 19 | 195.9 |
| Arequipa | 324 | 153 | −171 | −52.7 | 232 | −92 | −28.3 | 254 | −70 | −21.5 |
| Ayacucho | 51 | 48 | −3 | −6.3 | 61 | 10 | 19.1 | 56 | 5 | 9.4 |
| Cajamarca | 62 | 50 | −12 | −18.8 | 99 | 37 | 60.7 | 111 | 49 | 80.2 |
| Callao | 706 | 407 | −299 | −42.4 | 449 | −257 | −36.4 | 451 | −255 | −36.2 |
| Cusco | 120 | 140 | 20 | 16.9 | 198 | 78 | 65.3 | 220 | 100 | 83.6 |
| Huancavelica | 17 | 11 | −6 | −33.7 | 12 | −5 | −27.7 | 14 | −3 | −15.7 |
| Huanuco | 93 | 60 | −33 | −35.8 | 64 | −29 | −31.5 | 99 | 6 | 6.0 |
| Ica | 237 | 152 | −85 | −35.8 | 241 | 4 | 1.9 | 234 | −3 | −1.1 |
| Junin | 161 | 139 | −22 | −13.7 | 204 | 43 | 26.7 | 190 | 29 | 18.0 |
| La Libertad | 415 | 284 | −131 | −31.6 | 459 | 44 | 10.6 | 539 | 124 | 29.9 |
| Lambayeque | 254 | 211 | −43 | −17.0 | 380 | 126 | 49.5 | 354 | 100 | 39.3 |
| Lima | 3614 | 2721 | −893 | −24.7 | 3904 | 290 | 8.0 | 4282 | 668 | 18.5 |
| Loreto | 569 | 388 | −181 | −31.9 | 614 | 45 | 7.8 | 699 | 130 | 22.8 |
| Madre de Dios | 78 | 72 | −6 | −7.7 | 82 | 4 | 5.1 | 85 | 7 | 9.0 |
| Moquegua | 39 | 30 | −9 | −23.9 | 43 | 4 | 9.1 | 48 | 9 | 21.8 |
| Pasco | 8 | 14 | 6 | 70.7 | 15 | 7 | 82.9 | 23 | 15 | 180.5 |
| Piura | 257 | 252 | −5 | −1.8 | 357 | 100 | 39.1 | 421 | 164 | 64.1 |

*(Continued)*

| | Pre-pandemic (2015–19) | 2020 | | | 2021 | | | 2022 | | |
|---|---|---|---|---|---|---|---|---|---|---|
| | Mean (cases) | n | Differences in the number of cases | (%) | n | Differences in the number of cases | (%) | n | Differences in the number of cases | (%) |
| Puno | 64 | 62 | −2 | −3.1 | 104 | 40 | 62.5 | 115 | 51 | 79.7 |
| San Martin | 194 | 173 | −21 | −10.6 | 230 | 36 | 18.8 | 270 | 76 | 39.5 |
| Tacna | 102 | 65 | −37 | −36.0 | 76 | −26 | −25.2 | 100 | −2 | −1.6 |
| Tumbes | 69 | 77 | 8 | 11.0 | 62 | −7 | −10.7 | 126 | 57 | 81.6 |
| Ucayali | 337 | 289 | −48 | −14.2 | 435 | 98 | 29.1 | 450 | 113 | 33.5 |

lower in 2020, respectively (Table 3). In Callao region, the annual incidence of HIV was also 43% lower in 2021 (IRR = 0.57; 95% CI, 0.51–0.65) and 49% lower in 2022 (IRR = 0.51; 95% CI, 0.45–0.58), compared with the pre-pandemic period. In Amazonas region, the annual incidence of HIV was 35% lower in 2021 (IRR = 0.65; 95% CI, 0.52–080) increasing to more than 60% in 2022 (IRR = 1.60; 95% CI, 1.35–1.90), compared with the pre-pandemic period. Finally, in Madre de Dios region, the annual incidence of HIV was 14% lower in 2021 (IRR = 0.86; 95% CI, 0.62–1.19) and 15% lower in 2022 (IRR = 0.85; 95% CI, 0.62–1.17), compared with the pre-pandemic period (Table 3).

## 4. Discussion

Our findings shown a reduction in the incidence of HIV infection (26% reduction) in 2020 compared with the pre-pandemic period in Peruvian population. Although the literature is limited, some studies have reported a decrease in HIV incidence during the COVID-19 pandemic in 2020 in different countries [23,24]. During the COVID-19 pandemic, many outpatient services were limited or suspended to prevent the spread of SARS-CoV-2 and to conserve resources for COVID-19 care [3–5], including in Peru [25]. Consequently, laboratories had to focus on COVID-19 testing, resulting in a reduction in the availability of other diagnostic tests, particularly in the area of HIV testing and diagnosis (with reductions of up to 50% in HIV testing in some countries) [7,9,10]. In Peru, the average number of HIV diagnoses decreased by 33% by 2020 [19]. According to a recent study, the proportion of the Peruvian population tested for HIV in the past year decreased by an average of 8.33% (95% CI: −10.73% to −5.93%) [20]. Probably as a result of the decrease in HIV screening population, HIV incidence in the general population has decreased in 2020. In addition, our results showed a temporal association between the implementation of non-pharmaceutical interventions to reduce the spread of SARS-CoV-2 and the incidence of HIV infection in 2020. The COVID-19 pandemic confinement (the national lockdown) officially began on March 16, 2020. The strictest phase of the lockdown — where most businesses were closed, strict curfews were imposed, and only essential activities were allowed — lasted for about 3 and a half months, until the end of June 2020. After that, Peru gradually started a phased reopening, although restrictions and localized quarantines in specific regions continued for many more months. In response to the decline in the HIV infection rate, the Ministry of Health of Peru implemented several strategies during the COVID-19 pandemic to improve HIV testing and treatment services, particularly at the primary health care level. These measures include the decentralization of molecular testing, the introduction of contact tracing systems, and the decentralization of antiretroviral treatment [26]. We also observed that the incidence of HIV infection began to return to pre-pandemic rates in 2021 (with an increase of 5%), which was temporally associated with reducing or eliminating non-pharmaceutical interventions. The decrease in incidence could have several explanations. One possible explanation is that there was a decrease in transmission, that is, as a result of the non-pharmacological interventions, people may have decreased sexual and recreational activities. Another possible explanation is that infections continued to occur, but were less likely to be diagnosed because of the reduction in testing. However, there is not enough information to test these hypotheses. Some studies also showed that HIV screening declines by 2021 during the second or third COVID-19

**Table 2. HIV incidence by sex and age before and during COVID-19 pandemic in Peru.**

| | Pre-pandemic (2015−19) | | 2020 | | | 2021 | | | 2022 | | |
|---|---|---|---|---|---|---|---|---|---|---|---|
| | n | IR (95% CI) | n | IR (95% CI) | IRR (95% CI) | n | IR (95% CI) | IRR (95% CI) | n | IR (95% CI) | IRR (95% CI) |
| **Total** | 8141 | 25.03 (24.48-25.58) | 6025 | 18.47 (18.00-18.94) | 0.74 (0.71-0.76) | 8684 | 26.28 (25.73-26.84) | 1.05 (1.02-1.08) | 9750 | 29.19 (28.61-29.78) | 1.16 (1.13-1.20) |
| **Sex** | | | | | | | | | | | |
| Male | 6336 | 39.57 (38.60-40.55) | 4516 | 27.89 (27.10-28.70) | 0.71 (0.68-0.73) | 6895 | 41.20 (40.24-42.19) | 1.04 (1.01-1.08) | 7218 | 43.63 (42.63-44.65) | 1.10 (1.06-1.14) |
| Female | 1805 | 10.93 (10.43-11.45) | 1346 | 8.19 (7.76-8.64) | 0.75 (0.69-0.80) | 1635 | 10.00 (9.55-10.53) | 0.92 (0.86-0.98) | 1808 | 10.73 (10.20-11.20) | 0.98 (0.92-1.05) |
| **Age group (years)** | | | | | | | | | | | |
| 0–11 | 94 | 1.38 (1.11-1.69) | 35 | 0.54 (0.37-0.75) | 0.39 (0.26-0.58) | 43 | 0.70 (0.50-0.90) | 0.48 (0.33-0.70) | 80 | 1.25 (0.90-1.55) | 0.90 (0.66-1.23) |
| 12–17 | 162 | 4.65 (3.96-5.41) | 130 | 4.19 (3.50-4.98) | 0.90 (0.71-1.14) | 190 | 6.10 (5.23-6.98) | 1.30 (1.05-1.62) | 295 | 9.41 (8.36-10.54) | 2.03 (1.67-2.47) |
| 18–29 | 3659 | 54.15 (52.41-55.93) | 2758 | 42.94 (41.36-44.58) | 0.79 (0.75-0.83) | 3951 | 62.00 (60.07-63.95) | 1.14 (1.09-1.20) | 4026 | 63.67 (61.72-65.66) | 1.18 (1.12-1.23) |
| 30–59 | 3960 | 33.01 (31.99-34.05) | 2765 | 22.14 (21.32-22.98) | 0.67 (0.64-0.70) | 4093 | 32.10 (31.12-33.09) | 0.97 (0.93-1.01) | 4339 | 33.48 (32.49-34.49) | 1.01 (0.97-1.06) |
| ≥60 | 265 | 7.65 (6.75-8.62) | 174 | 4.20 (3.60-4.90) | 0.55 (0.45-0.67) | 253 | 5.90 (5.20-6.66) | 0.77 (0.64-0.92) | 286 | 6.48 (5.75-7.28) | 0.85 (0.71-1.01) |
| **Age group in men (years)*** | | | | | | | | | | | |
| 0–11 | 54 | 1.18 (0.85-1.61) | 19 | 0.58 (0.34-0.90) | 0.48 (0.27-0.86) | 26 | 0.80 (0.60-1.16) | 0.67 (0.39-1.12) | 51 | 1.57 (1.20-2.10) | 1.33 (0.86-2.05) |
| 12–17 | 96 | 6.40 (5.27-7.69) | 66 | 4.23 (3.30-5.40) | 0.66 (0.48-0.90) | 110 | 7.10 (5.80-8.50) | 1.10 (0.84-1.44) | 174 | 11.10 (9.53-12.90) | 1.74 (1.36-2.22) |
| 18–29 | 2942 | 115.69 (112.06-119.42) | 2171 | 70.43 (67.50-73.0) | 0.61 (0.57-0.64) | 3288 | 106.70 (103.06-110.40) | 0.92 (0.88-0.97) | 3339 | 109.70 (106.04-13.52) | 0.94 (0.91-1.00) |
| 30–59 | 3038 | 62.84 (60.82-64.91) | 2125 | 33.78 (32.40-35.60) | 0.54 (0.51-0.56) | 3272 | 52.00 (50.30-53.83) | 0.83 (0.79-0.86) | 3426 | 52.50 (50.78-54.31) | 0.84 (0.79-0.88) |
| ≥60 | 206 | 14.83 (13.03-16.81) | 135 | 6.84 (5.74-8.10) | 0.46 (0.31-0.57) | 199 | 10.10 (8.74-11.59) | 0.68 (0.56-0.82) | 228 | 10.90 (9.57-12.46) | 0.74 (0.61-0.89) |
| **Age group in women (years)*** | | | | | | | | | | | |
| 0–11 | 40 | 0.95 (0.65-1.35) | 16 | 0.50 (0.29-0.82) | 0.52 (0.26-0.98) | 17 | 0.53 (0.30-0.80) | 0.53 (0.28-0.99) | 29 | 0.92 (0.60-1.32) | 0.96 (0.56-1.63) |
| 12–17 | 67 | 4.76 (3.78-5.90) | 64 | 4.15 (3.20-5.30) | 0.87 (0.61-1.22) | 80 | 5.19 (4.12-6.46) | 1.09 (0.79-1.50) | 121 | 7.70 (6.39-9.20) | 1.62 (1.21-2.17) |
| 18–29 | 717 | 24.94 (23.30-26.66) | 587 | 17.57 (16.20-19.10) | 0.70 (0.63-0.78) | 663 | 19.85 (18.37-21.42) | 0.79 (0.72-0.88) | 687 | 20.94 (19.40-22.57) | 0.84 (0.75-0.93) |
| 30–59 | 922 | 16.21 (15.22-17.24) | 640 | 10.32 (9.54-11.15) | 0.63 (0.57-0.70) | 821 | 13.24 (12.35-14.18) | 0.82 (0.74-0.90) | 913 | 14.18 (13.28-15.13) | 0.87 (0.80-0.96) |
| ≥60 | 59 | 4.02 (3.15-5.05) | 39 | 1.80 (1.28-2.46) | 0.45 (0.29-0.67) | 54 | 2.49 (1.87-3.25) | 0.62 (0.43-0.89) | 58 | 2.49 (1.89-3.22) | 0.62 (0.43-0.87) |

Abbreviations: IR, incidence rate; IRR, incidence rate ratio; CI, confidence interval. *Population data were only available for 2019, 2020, 2021, and 2022 to calculate the incidence rate.

**Table 3. HIV incidence by region before and during COVID-19 pandemic in Peru.**

| | Pre-pandemic (2015–19) | | 2020 | | | 2021 | | | 2022 | | |
|---|---|---|---|---|---|---|---|---|---|---|---|
| Region | n | IR (95% CI) | n | IR (95% CI) | IRR (95% CI) | n | IR (95% CI) | IRR (95% CI) | n | IR (95% CI) | IRR (95% CI) |
| Amazonas | 221 | 51.85 (45.15-59.04) | 122 | 28.58 (23.70-34.10) | 0.55 (0.44-0.69) | 145 | 33.42 (28.20-39.30) | 0.65 (0.52-0.80) | 363 | 82.86 (74.60-91.80) | 1.60 (1.35-1.90) |
| Ancash | 137 | 11.73 (9.82-13.82) | 89 | 7.54 (6.10-9.30) | 0.64 (0.49-0.85) | 175 | 14.88 (12.80-17.30) | 1.27 (1.01-1.60) | 196 | 16.60 (14.40-19.10) | 1.42 (1.13-1.78) |
| Apurimac | 10 | 2.10 (1.03-3.94) | 10 | 2.32 (1.10-4.30) | 1.08 (0.40-2.90) | 25 | 5.89 (2.80-8.70) | 2.75 (1.28-6.42) | 27 | 6.38 (4.20-9.30) | 2.98 (1.40-6.90) |
| Arequipa | 324 | 23.96 (21.45-26.75) | 150 | 10.02 (8.50-11.70) | 0.42 (0.34-0.51) | 233 | 15.27 (22.40-17.40) | 0.64 (0.53-0.76) | 237 | 15.39 (13.50-17.50) | 0.64 (0.54-0.76) |
| Ayacucho | 51 | 7.12 (5.28-9.33) | 48 | 7.18 (5.30-9.50) | 1.01 (0.67-1.53) | 62 | 9.42 (7.20-12.10) | 1.34 (0.90-1.96) | 57 | 8.65 (6.50-11.20) | 0.93 (0.64-1.34) |
| Cajamarca | 62 | 3.99 (3.08-5.15) | 51 | 3.51 (2.60-4.60) | 0.87 (0.59-1.28) | 99 | 6.84 (4.60-8.30) | 1.70 (1.23-2.38) | 104 | 7.19 (5.90-8.70) | 1.79 (1.29-2.49) |
| Callao | 706 | 66.15 (61.33-71.18) | 396 | 35.05 (31.70-38.70) | 0.53 (0.47-0.60) | 434 | 37.82 (61.40-41.60) | 0.57 (0.51-0.65) | 394 | 34.02 (30.70-37.60) | 0.51 (0.45-0.58) |
| Cusco | 120 | 8.90 (7.39-10.66) | 139 | 10.24 (8.60-12.10) | 1.15 (0.89-1.48) | 196 | 14.41 (11.50-16.00) | 1.61 (1.28-2.05) | 217 | 15.90 (13.90-18.20) | 1.78 (1.42-2.25) |
| Huancavelica | 17 | 3.26 (1.95-5.35) | 11 | 3.01 (1.50-5.40) | 0.90 (0.38-2.04) | 13 | 3.68 (2.00-6.30) | 1.10 (0.49-2.41) | 14 | 4.02 (2.20-6.70) | 1.20 (0.55-2.59) |
| Huanuco | 93 | 10.50 (8.44-12.81) | 57 | 7.50 (5.70-9.70) | 0.72 (0.51-1.01) | 58 | 7.66 (9.80-9.90) | 0.73 (0.52-1.03) | 96 | 12.74 (10.30-15.60) | 1.22 (0.91-1.64) |
| Ica | 237 | 28.93 (25.40-32.91) | 147 | 15.07 (12.70-17.70) | 0.52 (0.42-0.64) | 219 | 22.27 (24.40-25.40) | 0.77 (0.64-0.93) | 201 | 20.09 (17.40-23.10) | 0.69 (0.57-0.84) |
| Junin | 161 | 11.58 (9.86-13.52) | 137 | 10.06 (8.50-11.90) | 0.87 (0.69-1.09) | 203 | 14.96 (13.90-17.20) | 1.29 (1.04-1.59) | 184 | 13.53 (11.60-15.60) | 1.17 (0.94-1.45) |
| La Libertad | 415 | 21.21 (19.22-23.35) | 280 | 13.88 (12.30-15.60) | 0.65 (0.56-0.76) | 449 | 22.10 (23.10-24.20) | 1.04 (0.91-1.19) | 495 | 24.10 (22.00-26.30) | 1.14 (0.99-1.30) |
| Lambayeque | 254 | 19.54 (17.20-22.08) | 195 | 14.88 (12.90-17.10) | 0.76 (0.63-0.92) | 372 | 28.02 (15.30-31.00) | 1.43 (1.22-1.69) | 342 | 25.56 (22.90-28.40) | 1.31 (1.11-1.54) |
| Lima | 3614 | 34.55 (33.44-35.70) | 2634 | 24.78 (23.80-25.70) | 0.72 (0.68-0.75) | 3835 | 35.16 (41.10-36.30) | 1.02 (0.97-1.06) | 3829 | 34.82 (33.70-35.90) | 1.01 (0.96-1.05) |
| Loreto | 569 | 52.83 (48.54-57.31) | 366 | 35.62 (32.10-39.50) | 0.67 (0.59-0.77) | 597 | 57.32 (63.80-62.10) | 1.08 (0.97-1.22) | 668 | 63.86 (59.10-68.90) | 0.93 (0.83-1.03) |
| Madre de Dios | 78 | 51.94 (41.05-64.82) | 73 | 42.00 (32.90-52.80) | 0.81 (0.58-1.13) | 82 | 44.77 (53.60-55.60) | 0.86 (0.62-1.19) | 83 | 44.13 (35.10-54.70) | 0.85 (0.62-1.17) |
| Moquegua | 39 | 20.96 (14.76-28.37) | 30 | 15.57 (10.50-22.00) | 0.75 (0.45-1.24) | 42 | 21.58 (18.50-29.20) | 1.04 (0.66-1.65) | 48 | 24.64 (18.20-32.70) | 1.19 (0.76-1.86) |
| Pasco | 8 | 2.62 (1.10-5.04) | 14 | 5.15 (2.81-8.64) | 2.01 (0.79-5.54) | 15 | 5.49 (3.07-9.06) | 2.15 (0.85-5.85) | 23 | 8.54 (5.41-12.81) | 3.34 (1.44-8.63) |
| Piura | 257 | 13.49 (11.91-15.27) | 249 | 12.16 (10.70-13.80) | 0.90 (0.75-1.07) | 353 | 16.93 (20.20-18.80) | 1.25 (1.06-1.48) | 419 | 19.87 (18.00-21.90) | 1.47 (1.26-1.72) |
| Puno | 64 | 4.35 (3.35-5.55) | 61 | 4.93 (3.80-6.30) | 1.13 (0.78-1.63) | 97 | 7.98 (4.50-9.70) | 1.57 (1.15-2.15) | 113 | 9.42 (7.80-11.30) | 2.16 (1.58-2.99) |
| San Martin | 194 | 21.88 (18.95-25.24) | 158 | 17.56 (14.90-20.50) | 0.80 (0.64-0.99) | 220 | 24.26 (23.20-27.70) | 1.84 (1.33-2.56) | 251 | 27.35 (24.10-31.00) | 1.24 (1.03-1.51) |
| Tacna | 102 | 28.36 (23.21-34.56) | 64 | 17.25 (13.30-22.00) | 0.61 (0.44-0.84) | 76 | 20.09 (33.80-25.10) | 0.71 (0.52-0.96) | 99 | 25.93 (21.10-31.60) | 0.91 (0.68-1.21) |
| Tumbes | 69 | 27.89 (21.57-35.09) | 78 | 31.01 (24.50-38.70) | 1.12 (0.80-1.57) | 60 | 23.40 (33.90-30.10) | 0.84 (0.59-1.21) | 125 | 48.16 (40.10-57.40) | 1.74 (1.28-2.36) |

*(Continued)*

**Table 3.** (Continued)

| Region | Pre-pandemic (2015−19) | | 2020 | | | 2021 | | | 2022 | | |
|---|---|---|---|---|---|---|---|---|---|---|---|
| | n | IR (95% CI) | n | IR (95% CI) | IRR (95% CI) | n | IR (95% CI) | IRR (95% CI) | n | IR (95% CI) | IRR (95% CI) |
| Ucayali | 337 | 65.03 (58.28-72.36) | 289 | 49.06 (43.60-55.00) | 0.75 (0.64-0.88) | 434 | 71.61 (62.00-78.70) | 1.10 (0.95-1.27) | 443 | 71.96 (65.40-78.90) | 1.11 (0.96-1.28) |

Abbreviations: IR, incidence rate; IRR, incidence rate ratio; CI, confidence interval.

pandemic wave [27,28]. By 2022, the incidence of HIV infection was higher than that during the pre-pandemic period. According to the literature, the Peruvian health system had partially or fully recovered its capacity to provide health care, as the number of users receiving care per month by December 2022 was similar to the pre-pandemic period [25].

Although HIV incidence decreased in both sexes, the annual incidence of HIV among men was 29% lower in 2020, compared with the pre-pandemic period, mainly for the 18–29 and 30–59 age groups, while the annual incidence in women was 25% lower. This change could be influenced by several factors. First, the pandemic caused a shift in health-care priorities, with resources being reallocated to COVID-19 efforts. This likely led to fewer HIV tests being conducted and, consequently, fewer diagnoses being recorded in men and in women [20], who are the population most affected by HIV infection. Second, lockdowns, social distancing measures, and restrictions on gatherings may have reduced opportunities for sexual activity with new partners, potentially decreasing HIV transmission rates among at-risk groups and men. Third, access to HIV prevention services, such as pre-exposure prophylaxis (PrEP), condom distribution, and education programs, may have been disrupted, affecting risk groups and men. Data from some studies show that PrEP use declined during the pandemic, likely resulting in fewer clinic-based PrEP follow-up visits [29,30]. This may have affected the trends observed, as reported in some countries [30–32]. Fourth, the economic and social disruptions of the pandemic could have altered behavior, including changes in migration patterns or access to healthcare, influencing HIV transmission dynamics among at-risk groups and men [33]. Compared with the pre-pandemic period, in 2021 (4% higher) and 2022 (10% higher) the annual incidence of HIV among men increased. These findings should be interpreted cautiously. It is likely that the decrease in incidence in the year 2020 and the increase in the years 2021 and 2022 reflect temporary changes due to the pandemic, in both men and women.

In 2020, 11 out of 25 Peruvian regions experienced a decline in annual HIV incidence compared to the pre-pandemic period. This decrease was most pronounced in regions with the highest pre-pandemic incidence rates, including Amazonas, Callao, Loreto, Madre de Dios, and Ucayali. Although there was a decreased in incidence of HIV infection, it is highly likely that HIV continued to spread in communities. HIV undiagnosis remains a significant public health problem in Peru; studies indicate that a significant number of people living with HIV are unaware of their status, challenging epidemic control, especially among at-risk groups [34–36]. As a concentrated epidemic, HIV is most prevalent in men who have sex with men, transgender women, and sex workers, where undiagnosed HIV infections may also be more common [35,36]. This poses a challenge to epidemic control. It is also important to note that the most pronounced decline occurred in the Amazonian regions of Peru (Loreto and Ucayali), where there is a considerable presence of indigenous populations. Preliminary evidence suggests that the conception and social representations of health and illness, especially in the case of HIV infection in indigenous populations, strongly influence the behaviors that they ultimately exhibit in response to health interventions [37,38]. Although the COVID-19 pandemic had receded by 2021 and 2022, the burden of HIV infection had increased in Peruvian regions with high prevalence of HIV infection. To address this problem, public health initiatives have been undertaken. The Centers for Disease Control and Prevention (CDC) and Peruvian health authorities have focused on outreach testing for key populations, index testing services (testing partners of HIV-infected persons), provider-initiated testing, and counseling. These approaches aim to increase diagnosed and reduce undiagnosed cases.

## Limitations

Our study has several limitations. First, our study was not able to address whether changes in health-seeking behavior, including a reluctance to visit health facilities (especially in 2020), may have contributed to the lower incidence of HIV infection during the COVID-19 pandemic. Second, only population data for the years 2019, 2020, 2021, and 2022 were available to estimate incidence by age group among males and females. Therefore, the lack of population data from 2015 to 2018 may have biased the estimates of expected incidence by age group by sex. It is also important to note that the HIV incidence rate varied slightly between 2015 and 2019, which was used to estimate the IR for the pre-pandemic period, and did not affect the IRR estimates for the pandemic period (years 2020, 2021, and 2022). Strengths of our study include five years of pre-pandemic surveillance, whereas similar surveillance data are scarce elsewhere in Latin America. As a result, our study provides the first and probably most comprehensive insights into the impact of the COVID-19 pandemic on HIV incidence rates and the burden of disease during the three years of the COVID-19 pandemic in Peru.

## Implications for Public Health

Our findings on HIV incidence change have several important public health implications for Peru. First, national estimates of changes in HIV incidence provide essential information for sexually transmitted infection (STI) surveillance, health services planning, and public health policy development. Second, these findings could serve as a basis for future STI control interventions and HIV screening initiatives. Third, our findings highlight the need for policies that can help minimize disruptions in HIV diagnosis and care during future health emergencies. In addition, the findings may inform the prioritization of control programs and the planning of STI-related health services in preparation for potential future public health crises.

## 5. Conclusions

Our study shows that during the COVID-19 pandemic in 2020, there was a decrease in the incidence of HIV infection in the Peruvian population. However, this incidence started to return to pre-pandemic rates as the COVID-19 pandemic receded, which coincided with the reduction or elimination of non-pharmaceutical interventions. By 2022, the incidence of HIV infection was higher than that of pre-pandemic levels, especially in regions of the Peruvian Amazon. The overall impact of reduced incidence could lead to delayed initiation of antiretroviral treatment and increased risk of transmission. Addressing this healthcare gap is essential to ensure that users who did not receive care during the pandemic have access to the healthcare services they need and, in addition, to mitigate long-term adverse health outcomes, especially among at-risk groups.

## Supporting information

**S1 Table. STROBE Statement—Checklist of items that should be included in reports of cross-sectional studies.**
(DOC)

**S2 Table. HIV incidence in Peru, 2015–2019.**
(DOC)

**S3 Table. HIV incidence in among the regions of Peru, 2015–2019.**
(DOC)

## Author contributions

**Conceptualization:** Max Carlos Ramírez-Soto.

**Data curation:** Max Carlos Ramírez-Soto, Hugo Arroyo-Hernández.

**Formal analysis:** Max Carlos Ramírez-Soto, Hugo Arroyo-Hernández.

**Investigation:** Max Carlos Ramírez-Soto, Hugo Arroyo-Hernández.

**Methodology:** Max Carlos Ramírez-Soto, Hugo Arroyo-Hernández.

**Software:** Max Carlos Ramírez-Soto, Hugo Arroyo-Hernández.

**Supervision:** Max Carlos Ramírez-Soto.

**Validation:** Max Carlos Ramírez-Soto, Hugo Arroyo-Hernández.

**Visualization:** Max Carlos Ramírez-Soto.

**Writing – original draft:** Max Carlos Ramírez-Soto, Hugo Arroyo-Hernández.

**Writing – review & editing:** Max Carlos Ramírez-Soto, Hugo Arroyo-Hernández.

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
