## [Decision Letter · Decision Letter 0]

10 Apr 2025

PONE-D-25-07972Changes in HIV incidence during the COVID-19 pandemic compared with pre-pandemic Peru: an observational studyPLOS ONE

Dear Dr. Ramírez-Soto,

Thank you for submitting your manuscript to PLOS ONE. After careful consideration, we feel that it has merit but does not fully meet PLOS ONE’s publication criteria as it currently stands. Therefore, we invite you to submit a revised version of the manuscript that addresses the points raised during the review process. I strongly suggest that the authors seek assistance with the revision of the English writing. While the overall content is clear, certain sections would beneBit from improved language Blow and grammar to ensure clarity and ease of reading fora broader audience.

The background of the study could be expanded. It would be helpful to include more details about the condition being studied, how it is typically diagnosed, and how the COVID-19 pandemic may have affected diagnosis rates. Furthermore, discussing the implications of missed diagnoses during the pandemic would strengthen the background and provide more context for the study.

The results section could be discussed in a more systematic and clearer manner. Currently, the presentation of the results feels somewhat fragmented. A more structured approach would enhance the readability and help the readers better understand the Bindings and their implications.

We look forward to receiving your revised manuscript.

Kind regards,

Oriana Rivera-Lozada de Bonilla

Academic Editor

PLOS ONE

Reviewer comments:

**Comments to the Author**

1. Is the manuscript technically sound, and do the data support the conclusions?

Reviewer #1: Partly

Reviewer #2: Partly

Reviewer #3: Yes

2. Has the statistical analysis been performed appropriately and rigorously? 

Reviewer #1: No

Reviewer #2: No

Reviewer #3: Yes

3. Have the authors made all data underlying the findings in their manuscript fully available?

Reviewer #1: Yes

Reviewer #2: Yes

Reviewer #3: No

4. Is the manuscript presented in an intelligible fashion and written in standard English?

Reviewer #1: Yes

Reviewer #2: Yes

Reviewer #3: No

5. Review Comments to the Author

Reviewer #1: 1. Study Design

The researchers used a cross-sectional study design but reported incidence rate ratios (IRRs).

Recommendation: They should either revise this approach or provide a clear justification for using IRRs in cross-sectional data.

2.Use of Simple Averages and Pooled Incidence Rates as a Baseline

The study averages HIV cases from 2015–2019 to establish a pre-pandemic reference and pools incidence rates over these years. This assumes that HIV incidence was stable, which may not be true. HIV cases could have been rising or falling before the pandemic due to policy changes, testing rates, or other factors. Simply averaging or pooling data ignores year-to-year variations.

Recommendations:

a. Instead of a simple average, the researchers may use time-series methods to estimate the expected trend had the pandemic not occurred.

b. The researchers may calculate annual incidence rates (IRs) and test for a linear trend using regression models. This will show if HIV incidence was already changing before COVID-19.

3. Discussion and Future Research

The researchers could expand the discussion on the implications of their findings and suggest future research directions.

4. The phrase “By 2022.” on page 2, line 33 needs clarification.

Reviewer #2: Many thanks for the opportunity to review this article. I think it is a valuable piece of work as it is very important to well describe what the impact of the non-pharmaceutical interventions was on the health status of the populations in different countries and regions of the world during the Covid 19 pandemic. I feel the paper still needs some significant work before being published. I have provided many comments, questions and suggestions to help the authors improve the article. I hope these help. To me the main limitation, that should be addressed, is that the study is considering the average new infection numbers and IR for the years 2015 to 2019 . Some of the decreases described for 2020 may have started in 2018 or 2019 and could be explained by other factors. We cannot know this if we do not see the trends from 2015 to 2019. To draw strong conclusions, more robust approaches should be used, e.g. time series analyses. I am aware that these analyses are not always possible, and in some occasions a simpler approach can be enough to demonstrate a given hypothesis or to highlight a fact. In any case, I would at least present somewhere yearly trends for 2015 to 2019. And I would also work in all the other aspects that I commented about.

Reviewer #3: Thank you for the opportunity to review your manuscript.

I appreciate the effort and the valuable insights presented. However, I have a few points I would like to comment on. In addition to the detailed revisions, I would like to highlight three general observations.

Please see attachment.

6. PLOS authors have the option to publish the peer review history of their article (what does this mean? ). If published, this will include your full peer review and any attached files.

**Do you want your identity to be public for this peer review?** For information about this choice, including consent withdrawal, please see our Privacy Policy .

Reviewer #1: No

Reviewer #2: **Yes: ** Antonio Isidro Carrion Martin

Reviewer #3: No

---

## [Author Response · Author response to Decision Letter 1]

29 Apr 2025

Response to reviewers

Title of the article: Changes in HIV incidence during the COVID-19 pandemic compared with pre-pandemic Peru: an observational study

Ms. Ref. No.: PONE-D-25-07972

Dear Editor,

Thank you for your comments and those of the reviewers. We believe our manuscript has improved considerably.

Reviewer #1

Comment 1. Study Design The researchers used a cross-sectional study design but reported incidence rate ratios (IRRs). Recommendation: They should either revise this approach or provide a clear justification for using IRRs in cross-sectional data.

Response 1. Thank you for your comment. This is an observational study using surveillance data from the national health system. Therefore, we have corrected the design only as an observational study of health system surveillance.

Comment 2. Use of Simple Averages and Pooled Incidence Rates as a Baseline The study averages HIV cases from 2015–2019 to establish a pre-pandemic reference and pools incidence rates over these years. This assumes that HIV incidence was stable, which may not be true. HIV cases could have been rising or falling before the pandemic due to policy changes, testing rates, or other factors. Simply averaging or pooling data ignores year-to-year variations. Recommendations:

a. Instead of a simple average, the researchers may use time-series methods to estimate the expected trend had the pandemic not occurred.

b. The researchers may calculate annual incidence rates (IRs) and test for a linear trend using regression models. This will show if HIV incidence was already changing before COVID-19.

Response 2. Thank you for your comment. Thank you for your detailed comments, they help us to improve the study. A time series study would require a modeled analysis and other data, which we will consider in a future study. Following your recommendations, we have included trends in overall and regional incidence rates with 95% confidence intervals (see supplementary material, Table S2). These show slight changes in incidence rates that do not affect the study findings.

Comment 3. Discussion and Future Research The researchers could expand the discussion on the implications of their findings and suggest future research directions.

Response 3. Thank you for your comment. We have included an “Implications for Public Health” paragraph in the Discussion section.

Comment 4. The phrase “By 2022.” on page 2, line 33 needs clarification.

Response 4. Thank you for your comment. The error has been corrected.

Reviewer #2

Many thanks for the opportunity to review this article. I think it is a valuable piece of work as it is very important to well describe what the impact of the non-pharmaceutical interventions was on the health status of the populations in different countries and regions of the world during the Covid 19 pandemic. I feel the paper still needs some significant work before being published. I have provided many comments, questions and suggestions to help the authors improve the article. I hope these help. To me the main limitation, that should be addressed, is that the study is considering the average new infection numbers and IR for the years 2015 to 2019. Some of the decreases described for 2020 may have started in 2018 or 2019 and could be explained by other factors. We cannot know this if we do not see the trends from 2015 to 2019. To draw strong conclusions, more robust approaches should be used, e.g. time series analyses. I am aware that these analyses are not always possible, and in some occasions a simpler approach can be enough to demonstrate a given hypothesis or to highlight a fact. In any case, I would at least present somewhere yearly trends for 2015 to 2019. And I would also work in all the other aspects that I commented about.

Response 1. Thank you for your comment. Corregido. Hemos incluido las tendencias en las tasas de incidencia general y por regiones con intervalos de confianza del 95% (ver material suplementario, Table S2). Donde se muestras ligeros cambios en las tasas de incidencia que no afectan los hallazgos del estudio.

Introduction

Response 2. Thank you for your comment. We have corrected all spelling and grammatical errors in the new version of the manuscript.

Methods

We have corrected all spelling and grammatical errors in the new version of the manuscript.

Comment 2. average number per year? or rather IR?

Response 2. Thank you for your comment. The statistical analysis section also describes the calculation of the RI. This value is the number of cases estimated for the pre-pandemic period. The incidence rate is described in other paragraph in the Methods section.

Comment 3. how were they calculated? Poisson regression?

Response 3. Thank you for your comment. We have included some clarifications about the IRR.

Results

We have corrected all spelling and grammatical errors in the new version of the manuscript.

Comment 1. I would only use one decimal digit (if needed at all), but whatever is chosen, I would be consistent, in the tables some figures have two decimals others only one

Response 1. Thank you for your comment. The decimals in tables 2 and 3 have been corrected. We use two decimals.

Discussion

We have corrected all spelling and grammatical errors in the new version of the manuscript.

Comment 1. I have no doubt that this is true, but there has not been any details or dates given for NPI in Peru, perhaps the authors should explain more? e.g. when lock down started, duration, when reduction in HIV clinics started, etc.

Response 1. Thank you for your comment. We have included a brief explanation of confinement in Peru.

Comment 2. To me the big question would be: was there a decrease in IR because of an actual reduction in transmission, i..e, people decrease sexual, recreational activities, etc (as a consequence of the NPIs) pr was the decrease rather a result on of the decrease in testing (i.e. infections continued to occur but were less likely to be diagnosed). I guess it was a bit of both? In any case I wonder if this IRs higher than the pre pandemic levels could back the hypothesis that IR decrease was a result of lack of testing, so a "surveillance artifact" , i.e. spread continued at similar levels and eventually even higher levels as a consequence of the increase of undiagnosed people.. In any case I feel this is not discussed enough..

Response 2. Thank you for your comment. We have included a brief explanation of the possible transmission of HIV.

Comment 3. was there further decrease in testing? did testing start to increase gain in 2021?

Response 3. Thank you for your comment. The sentence has been removed to avoid reader confusion. In addition. Paragraph 3 of this section mentions the decline in HIV testing.

Comment 4. Not clear what this means, "able to care for several patients"? I guess you mean that your findings suggest a recovery of the health system ... which has also been documented in other studies? or something like that?

Response 4. Thank you for your comment. We've corrected the sentence to explain the changes in healthcare in Peru.

Comment 5. but proportional reduction was higher in females, right? I think that if you decide to focus on males because they represent a higher % of infections, you should somehow explain this.

Response 5. Thank you for your comment. . It has been explained that both sexes were affected, including those at risk.

Comment 6. The several factors would apply to overall changes, not only in men, right? To me those factors are the most important part of the discussion, and it is not clear why the focus only on men, i would use these factors to explain overall results and then maybe say something about males?

Response 6. Thank you for your comment. It has been explained that both sexes were affected, including those at risk.

Comment 7. Include the reference “Fourth, the economic and social disruptions of the pandemic could have altered behavior, including changes in migration patterns or access to healthcare, influencing HIV transmission dynamics”.

Response 7. Thank you for your comment. The reference has been included “33. Winwood JJ, Fitzgerald L, Gardiner B, Hannan K, Howard C, Mutch A. Exploring the Social Impacts of the COVID-19 Pandemic on People Living with HIV (PLHIV): A Scoping Review. AIDS Behav. 2021;25(12):4125-4140. doi: 10.1007/s10461-021-03300-1”.

Comment 8. “The observed decline in HIV incidence during this period should be interpreted with caution. It is likely that the reduced incidence reflects temporary changes due to the pandemic”. But this is for the overall figures, not only for men, right?

Response 8. Thank you for your comment. It has been explained that men and women were affected.

Comment 9. I feel there is a disconnection between this section and the precedent phrases, do you need to explain all the results from these papers, could you not just say that the problem of diagnosed infections may be more prounounced in those specific groups, and reference the papers.

Response 9. Thank you for your comment. The sentence has been corrected according to your recommendations.

Comment 10. “Prior to the pandemic, a study of men who have sex with men and 255 transgender women in the Lima region found that 31% were infected with HIV, of whom 35% had not been previously diagnosed [35]. Another study conducted in the Cajamarca region between 2015 and 2021 estimated that of 1,388 people living with HIV, approximately 393 cases (28.4%) were undiagnosed [36]”. I feel there is a disconnection between this section and the precedent phrases, do you need to explain all the results from these papers, could you not just say that the problem of diagnosed infections may be more prounounced in those specific groups, and reference the papers.

Response 10. Thank you for your comment. The paragraph has been rewritten for better understanding of undiagnosed cases.

Comment 11. “Another group considered vulnerable”. This feels like is not really related to the paper.. I would introduce this differently, and say that the most pronounced decrease occurred in regions with an important presence of indigenous populations... this could be explained by ...

Response 11. Thank you for your comment. We have corrected the sentence according to your recommendations.

Comment 12. Are these the same studies as 37 and 38? do they refer only to indigenous population? not clear.. if you are not talking about indigenous anymore, perhaps move into different paragraph?

Response 12. Thank you for your comment. The information has been reorganized for better understanding by readers.

Comment 13. the actions described were already put in place a while ago, before this paper. Any new public health recommendation from your own results?

Response 13. Thank you for your comment. We have included a paragraph with the implications for public health.

Comment 14. One of the big limitations of the study is that it considers average numbers and IR for the years 2015 to 2019, some of the decreases described for 2020 may have started in 2018 or 2019 and could be also explained by other factors. To draw strong conclusions more robust approaches should be used, e.g time series analyses. Although at times these analyses are not possible. In any case, I would at least present somewhere yearly trends for 1025 to 2019.

Response 14. Thank you for your comment. We have included a brief description of the IR estimates that do not affect the estimates in the limitations section.

Comment 15. Conclusiones. I guess you mean the overall reduction in diagnosis could have led... and then you say about deaths, but I do not think these are conclusions from your study? I would remove from conclusions.

Response 15. Thank you for your comment. The sentence has been deleted.

Reviewer #3

Thank you for the opportunity to review your manuscript. I appreciate the effort and the valuable insights presented. However, I have a few points I would like to comment on. In addition to the detailed revisions, I would like to highlight three general observations:

• English Writing Support: I strongly suggest that the authors seek assistance with the revision of the English writing. While the overall content is clear, certain sections would beneBit from improved language Blow and grammar to ensure clarity and ease of reading for a broader audience.

• Background of the Study: The background of the study could be expanded. It would be helpful to include more details about the condition being studied, how it is typically diagnosed, and how the COVID-19 pandemic may have affected diagnosis rates. Furthermore, discussing the implications of missed diagnoses during the pandemic would strengthen the background and provide more context for the study.

• Clarity and Systematization of Results: The results section could be discussed in a more systematic and clearer manner. Currently, the presentation of the results feels somewhat fragmented. A more structured approach would enhance the readability and help the readers better understand the Bindings and their implications.

ABSTRACT

Comment 1. LINES 26-29. There are a couple of issues in this sentence that need correction. First, the phrase "from 25 geographically diverse from Peru" is unclear and likely missing a noun after "diverse". A suggested revision is "from 25 geographically diverse regions in Peru". Second, there is a subject-verb agreement error in "HIV incidence during the COVID-19 pandemic years (2020, 2021, and 2022) were compared". Since HIV incidence is singular, the verb should also be singular: "was compared". Please revise accordingly.

Response 1. Thank you for your comment. We have corrected the sentence according to your recommendations.

Comment 2. LINE 33. The words “By 2022” seem out of place.

Response 2. Thank you for your comment. The error has been corrected.

Comment 3. LINE 35. Consider removing the “and” before 4% for smoother phrasing.

Response 3. Thank you for your comment. The error has been corrected.

Comment 4. LINE 36. Age stratified should be age-stratified.

Response 4. Thank you for your comment. The error has been corrected.

Comment 5. LINE 42. “Was” should be “has” to reflect the appropriate tense.

Response 5. Thank you for your comment. The error has been corrected.

Comment 6. LINES 44-45 The sentence is grammatically incorrect. It should be: “By 2022, the incidence of HIV infection was higher than in the pre-pandemic period, especially in regions of the Peruvian Amazon."

Response 6. Thank you for your comment. We have corrected the sentence according to your recommendations.

INTRODUCTION

Comment 1. LINE 51. The authors mention non-pharmaceutical interventions but never explain what they are. Although this is not the main focus of the work, I think providing a brief explanation would improve the Blow of the text.

Response 1. Thank you for your comment. A description of non-pharmacological interventions has been included.

Comment 2. LINE 73. The sentence “In China, a 59% reduction in testing across four regions” is incomplete and lacks a main verb. Consider revising it to something like, “In China, a 59% reduction in testing was observed across four regions.”

Response 2. Thank you for your comment. We have corrected the sentence according to your recommendations.

Comment 3. LINES 75-76 The phrase "a more than 50% drop" sounds slightly unnatural. A clearer construction would be "more than a 50% drop" or "a drop of more than 50%". Please consider revising for better readability.

Response 3. Thank you for your comment. We have corrected the sentence according to your recommendations.

Comment 4. LINES 77-78. The authors state that two studies assessed changes in HIV diagnoses in Peru but only present the results from one of them. Consider addressing why the second study's results are not presented or clarify the reasoning behind this.

Response 4. Thank you for your comment. The results of another study were included.

Comment 5. LINES 84-85. I’m not sure if “changes” is the appropriate term for conveying that undiagnosed HIV cases have implications for treatment initiation. Consider using "delays" or "barriers" instead, if that is the intended meaning.

Response 5. Thank you for your comment. We have corrected

---

## [Editor Report · Decision Letter 1]

2 May 2025

Changes in HIV incidence during the COVID-19 pandemic (2020–22) compared with the pre-pandemic period (2015–19) in Peru: an observational study

PONE-D-25-07972R1

Dear Dr. Max Carlos Ramírez-Soto,

We’re pleased to inform you that your manuscript has been judged scientifically suitable for publication and will be formally accepted for publication once it meets all outstanding technical requirements.

Kind regards,

Oriana Rivera-Lozada de Bonilla

Academic Editor

PLOS ONE

---

## [Editor Report · Acceptance letter]

PONE-D-25-07972R1

PLOS ONE

Dear Dr. Ramírez-Soto,

I'm pleased to inform you that your manuscript has been deemed suitable for publication in PLOS ONE. Congratulations! Your manuscript is now being handed over to our production team.

Kind regards,

on behalf of

Dr. Oriana Rivera-Lozada de Bonilla

Academic Editor

PLOS ONE